# On the Effectiveness of Fine-tuning Versus Meta-reinforcement Learning

**Zhao Mandi**  **Pieter Abbeel**  **Stephen James**

{mandi.zhao, pabbeel, stepjam}@berkeley.edu
University of California, Berkeley

## Abstract

Intelligent agents should have the ability to leverage knowledge from previously learned tasks in order to learn new ones quickly and efficiently. Meta-learning approaches have emerged as a popular solution to achieve this. However, meta-reinforcement learning (meta-RL) algorithms have thus far been predominately validated on simple environments with narrow task distributions. Moreover, the paradigm of pretraining followed by fine-tuning to adapt to new tasks has emerged as a simple yet effective solution in supervised and self-supervised learning. This calls into question the benefits of meta-learning approaches in reinforcement learning, which typically come at the cost of high complexity. We therefore investigate meta-RL approaches in a variety of vision-based benchmarks, including Procgen, RLBench, and Atari, where evaluations are made on completely novel tasks. Our findings show that when meta-learning approaches are evaluated on different tasks (rather than different variations of the same task), multi-task pretraining with fine-tuning on new tasks performs equally as well, or better, than meta-pretraining with meta test-time adaptation. This is encouraging for future research, as multi-task pretraining tends to be simpler and computationally cheaper than meta-RL. From these findings, we advocate for evaluating future meta-RL methods on more challenging tasks, and including multi-task pretraining with fine-tuning as a simple yet strong baseline. [1]

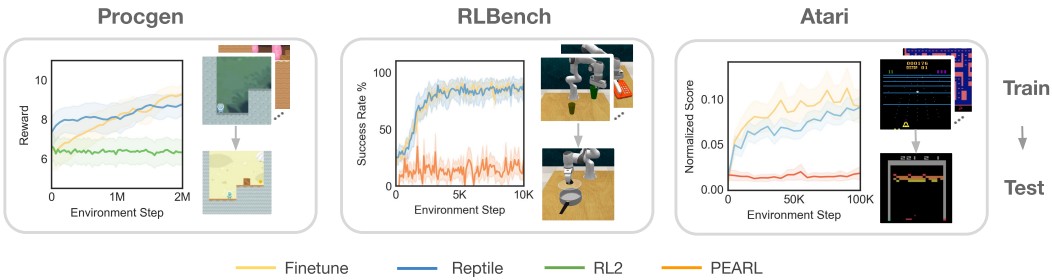

Figure 1: Meta reinforcement learning (meta-RL) algorithms have predominantly been validated on simple environments with narrow task distributions. Our study investigates meta-RL algorithms on much wider task distributions across 3 benchmarks, and concludes that **multi-task pretraining with fine-tuning on new tasks performs equally as well, or better, than meta-RL.** Each of the three plots shows the average performance across tasks in the corresponding benchmark.

---

[1]Project Website: https://sites.google.com/berkeley.edu/finetune-vs-metarl

36th Conference on Neural Information Processing Systems (NeurIPS 2022).

# 1   Introduction

One of the major gaps between human and machine intelligence is the sample efficiency of learning. Whereas humans can leverage past knowledge to learn a new task from a few examples, current machine learning systems often require a large amount of data and supervision to achieve even a single task. To bridge this gap, meta-learning has become a popular approach — it uses many tasks to meta-train an optimal learning strategy, which enables few-shot generalization on a test task.

Meta-learning methods have had the most success in supervised learning settings [1–3], specifically few-shot image classification, where the goal is to learn a classifier to recognize unseen classes during a test-time training phase with limited labeled data. Recent work has found that variations of simple pretraining and fine-tuning can perform equally as well as more complex meta-learning approaches [4–7]. However, within the realm of reinforcement learning, simple pretraining and fine-tuning is not known to out perform meta-reinforcement learning (meta-RL). Our hypothesis for this discrepancy is simple: the computer vision (CV) community evaluates their approaches on distinct test tasks (e.g. classifying dogs, cats, and birds), while the meta-RL community evaluates on *variations* of the same train-time tasks; for example, varying transition dynamics (e.g. different friction parameters) or varying reward functions (e.g. running forward v.s. running backward) are better categorized as variations rather than different tasks, as discussed in recent work [8, 9]. *Variation* adaptation is inherently easier than *task* adaptation, and does not paint a full picture of the shortcomings of meta-RL. Moreover, most meta-RL methods (with a few exceptions, discussed in 7) have been studied in fully observable settings or with shaped rewards [10–12], neglecting more realistic real-world scenarios, where rewards are often sparse, and observations are high-dimensional (e.g. images, point-clouds, etc).

Evidently, there is a gap in the literature, where a large-scale study would be well placed to analyze the setting of vision-based meta-RL across a *truly* diverse set of tasks. We use 3 existing RL benchmarks: Procgen [13], RLBench [8], and Atari Learning Environment (ALE) [14, 15] — each offers a diverse set of distinct tasks that we use for train and test. For example in RLBench [8], an agent could be trained to pick up cups, take a USB out of a computer, and reach target locations, while at test time, adaptation would be evaluated on completely unseen tasks, such as lifting blocks and pushing buttons.

We investigate three prominent meta-RL algorithms of differing paradigms: Reptile [11] — a gradient-based method, PEARL [12] — a context-based method, and $RL^2$ [16] — an LSTM-based method. Results from this study are enlightening: multi-task pretraining, followed by fine-tuning on novel tasks, performs equally as well, or better, than all three meta-RL algorithms, while being much simpler and less computationally expensive to train. In light of this, we advocate for future research to shift towards more challenging benchmarks, and include multi-task pretraining with fine-tuning as a simple, yet strong baseline.

To summarize, the key **contributions** of our study are as follows:
- We show that multi-task pretraining followed by fine-tuning on novel tasks performs equally as well, or better, than common meta-RL baselines on vision-based environments.
- We present the first large-scale multi-task and meta-RL study on three existing benchmarks: cross-level adaptation on Procgen, cross-task adaptation on RLBench, and cross-game adaptation on Atari.
- Within RLBench, we show that large-scale multi-task pretraining can overcome sparse rewards on unseen test tasks and perform significantly better than training from scratch .

# 2   Preliminaries

**Reinforcement Learning.**   Reinforcement learning (RL) assumes access to an agent that interacts with an environment in which there are states $\mathbf{s} \in \mathcal{S}$, actions $\mathbf{a} \in \mathcal{A}$, and a reward function $R(\mathbf{s}_t, \mathbf{a}_t)$, where $t$ represents the current time step. The agent must discover a policy $\pi$ that maximizes the expectation of the sum of discounted rewards, i.e. $\mathbb{E}_\pi[\sum_t \gamma^t R(\mathbf{s}_t, \mathbf{a}_t)]$, where future rewards are discounted by a factor $\gamma \in [0, 1)$. Each policy $\pi$ has a corresponding Q-value function $Q(\mathbf{s}, \mathbf{a})$, which defines the agents expected return when following the policy after taking action $\mathbf{a}$ in state $\mathbf{s}$.

**Meta-Reinforcement Learning.**   Meta-reinforcement learning (meta-RL) aims to quickly adapt to any task via RL. Given a distribution of tasks $p(\mathcal{T})$, where each sampled task $\mathcal{T}_i \in p(\mathcal{T})$, is a stand-alone Markov Decision Process (MDP) $\mathcal{T}_i = (\mathcal{S}, \mathcal{A}, P, \gamma, R)_i$ or a partially observable MDP

(POMDP) $\mathcal{T}_i = (\mathcal{S}, \mathcal{A}, P, \gamma, R, \mathcal{O})_i$, in the case where an agent receives observations $\mathbf{o} \in \mathcal{O}$ rather than ground-truth states $\mathbf{s} \in \mathcal{S}$. Given a task $\mathcal{T}$, the agent is allowed to collect a small amount of data $\mathcal{D}_\mathcal{T}$ and use it to adapt the policy to obtain $\pi_\mathcal{T}$ with an RL algorithm. Meta-RL training aims to find an optimal initial policy or exploration strategy that maximizes the expected return of its adapted policies across all tasks within a limited number of steps: $\mathbb{E}_{\mathcal{T} \sim p(\mathcal{T}), (\mathbf{s}_t, \mathbf{a}_t) \sim \pi_\mathcal{T}}[\sum_t \gamma^t R(\mathbf{s}_t, \mathbf{a}_t)]$.

**Assumption on Task Distribution.** We assume no ambiguity exists between any two tasks in the train or test sets. Namely, all tasks have distinct, non-overlapping state-action spaces: $\forall \mathcal{T}_i, \mathcal{T}_j \in p(\mathcal{T}), i \neq j, \mathcal{S}_i \times \mathcal{A}_i \cap \mathcal{S}_j \times \mathcal{A}_j = \varnothing$. An agent should be able to infer which task to perform based on a single-step observation (e.g. whether the current task is playing Breakout or Pong is always immediately distinguishable from a single observation). As a result, our multi-task pretraining baseline does not need a separate indicator to determine which task to complete. We remark that our assumption is different from most prior work in meta-RL, which often assumes the difference in "tasks" only occurs in transition dynamics (e.g. two tasks are both half-cheetah running, but with different friction parameters) or the reward function (e.g. running forward v.s. running backward) — we therefore categorize such assumptions as "variation adaptation", and not "task adaptation", to be better distinguished from our work. Please see Figure 2 for illustration.

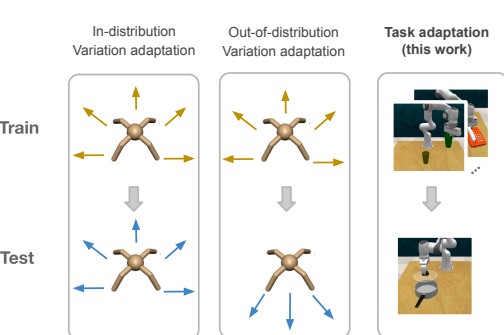

Figure 2: An illustrative comparison between the task assumptions from prior works' (in-distribution or out-of-distribution adaptation) and from ours (task adaptation). In our assumption, we emphasize both the high diversity of training tasks — which should lead to generalizable skills, and strictly unseen test-time tasks — which poses a more challenging adaptation problem than simple retrieval of trained skills.

### 2.1 Overview of Compared Algorithms

**Reptile [11]** is a first-order, gradient-based meta-learning algorithm. It finds initial model parameters that can be quickly adapted to any task via gradient descent. Meta-training alternates between an inner loop — which performs multiple gradient steps on a single task, and an outer loop — which mixes the updated parameters from multiple tasks. Reptile is mathematically similar to first-order model-agnostic meta-learning [3] (MAML) while being easier to implement, and achieves comparable results as MAML in both supervised [11] and imitation learning [17] settings. We choose Reptile for its simplicity and applicability to both on-policy and off-policy RL algorithms.

**PEARL [12]** is a context-based, off-policy meta-RL algorithm. It meta-trains a policy that conditions on context variables, which are encoded from small amounts of an agent's experiences and are used to perform online probabilistic filtering to infer how to solve a new task. At test time, it adapts to a new task by alternating between 1) collecting new experiences to update the context and 2) adjusting its policy according to the updated context.

**RL$^2$ [16]** is a recurrence-based, policy-gradient meta-RL algorithm. It represents the RL policy with a recurrent neural network (RNN) that embeds agent experiences from multiple "trials" in a single task, with the aim of encoding an RL update rule into the RNN weights. At test time, it adapts by rolling out an agent in a new task, which updates the RNN hidden states and improves the policy to success after a few trials.

**Multi-task Training** As a baseline, we run multi-task reinforcement learning (MTRL), which simply learns to achieve all the training tasks simultaneously with a single RL policy. At test time, the agent adapts by finetuning the policy on a new task using the same RL algorithm as training.

**Base RL Algorithms** Each of the above methods is applied to a base RL algorithm that are different across benchmarks. For our Progen experiments (Section 3), we use PPO [18]. For our RLBench experiments (Section 5), we use C2F-ARM [19]. For our Atari experiments (Section 7), we use RainbowDQN [20] as the base algorithm. We choose the use the base algorithm that is commonly used and well-tuned for each environment, and provide more detailed justifications for each choice

in their respective sections. See Algorithms 1 and 2 in Section 6 for a unified overview of how the meta-RL algorithms are implemented.

All experiments in the study are trained and evaluated on a maximum of 8 RTX A5000 GPUs, each with 24GB of memory. Unless otherwise indicated, we use the same hyperparameters and architectures as used in prior work; full details are given in the appendix.

# 3 Procgen Experiments

Procgen Benchmark [13] is a suite of procedurally-generated game-like environments designed for studying generalization in deep RL, where an agent is typically trained simultaneously on a set of levels and evaluated on unseen levels. We rank Procgen as the easiest benchmark in this study, because both the embodiment and task objects are consistent across tasks, while only the colors and layouts vary.

Prior work has explored improving generalization on Procgen by training on a large amount of levels [13], adding data-augmentation to visual inputs [21, 22], knowledge transfer during training [23], and self-supervised world models [24]. However, beyond zero-shot generalization, few prior works use Procgen to evaluate adaptation in meta-RL, with one exception being *Alver et al.* [25], which showed that RL$^2$ failed to generalize on simplified Procgen games.

We choose one representative Procgen game, namely Coinrun, where an agent is tasked to avoid obstacles and reach a golden coin. The game contains multiple "levels", each of which has a unique color theme or layout. We use up to 10,000 levels for training, and held-out 20 levels for testing. 100 million environment steps are used for all pretraining runs, which amounts to 1500 PPO iterations. Then the final model checkpoint is used to test adaptation — the environment is run in parallel on 256 threads and each PPO iteration uses 256 steps.

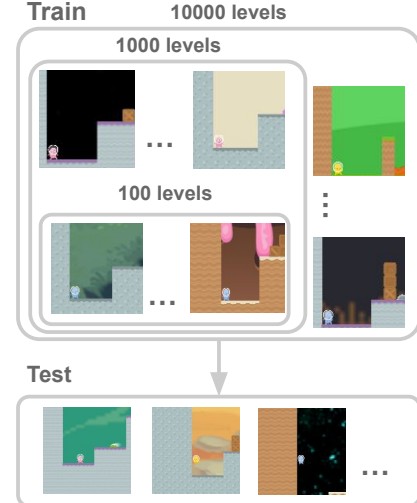

Figure 3: In Procgen experiments, each agent is trained on various number of Coinrun levels and tested on 20 unseen levels.

## 3.1 Training Setup

We use PPO [18] as the base RL algorithm and use the same image encoder as IMPALA [26], which has been shown to achieve competitive performance on single-game settings in Procgen [13, 27]. Note that PEARL is excluded from this section because it is based on off-policy RL algorithms.

**Reptile-PPO** incorporates the inner-outer loop in Reptile [11] with PPO. For each iteration in the inner loop, the environment is fixed to one randomly sampled task level, where rollouts are collected and used to perform $k$ iterations of batched gradient updates (we use $k = 3$ in our experiments). Following that, the outer loop takes the updated policy parameters, and performs a soft parameter update. This process is repeated iteratively.

**RL$^2$-PPO** combines an LSTM with PPO as proposed in [16]. We modify the Coinrun environment to re-sample the same level for multiple episodes, such that one original environment episode is treated as one "trial", and each RL$^2$ episode contains multiple "trials". Until each multi-trial episode is terminated, the agent's past rollouts are concatenated and fed into an LSTM layer. The image observations are first encoded by the IMPALA (following [13]) and flattened into 256-dimensional latent embeddings, then the latent embeddings are concatenated with action, reward, and done signals. The concatenated vectors are embedded again before being fed into LSTM hidden layers.

**MT-PPO** jointly trains PPO on multiple levels of Coinrun. The environment randomly samples from a fixed number of training levels, and the policy is updated with a mixture of rollouts that are collected from multiple environment levels.

## 3.2 Test-time Adaptation Setup

For **Reptile-PPO** and **MT-PPO**, a trained agent is finetuned with vanilla PPO for 2 million environment steps on each test level. For $RL^2$**-PPO**, the LSTM hidden states are reset, and the agent is rolled-out on each new level for 2 million environment steps without gradient updates. We also compare adaptation with training from scratch, where one agent is trained for each test level with PPO for 2 million steps. We use a total of 20 held-out levels for testing, and results are reported by averaging across all 20 levels.

## 3.3 Results

Results for testing on unseen levels are shown in Figure 4. We first remark that **finetuning achieves the best performance at adaptation to new levels** both in terms of sample efficiency and final performance, and is shown consistently across varying number of training levels. Notably, Reptile-PPO is also able to improve performance on test tasks, but the finetuning performance is worse than MT-PPO, which suggests the Reptile-learned parameters provides a less adaptable initialization than the simpler multi-task trained parameters. Figure 4 also shows that increasing the number of training levels improves zero-shot performance on MT-PPO (and less so for Reptile-PPO), which is consistent

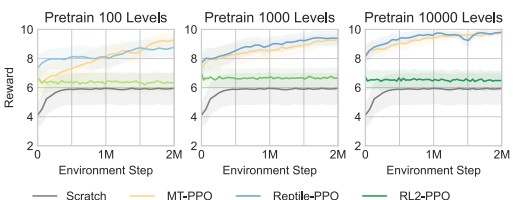

Figure 4: pretrained Procgen(Coinrun) agents on unseen levels. Each sub-figure corresponds to pretraining on a different number of levels, ranging from 100, 1000, 10000. Solid lines are average over 5 seeds, with shaded regions representing standard errors.

with the aforementioned prior work that studys generalization on Procgen. Expanding the training set also benefits finetuning and provides a clear advantage over training from scratch, which is encouraging news for simple scaling of training tasks as an alternative to designing computationally complex meta-training algorithms.

$RL^2$ does not improve performance on unseen levels, meaning that the updated hidden state of the RNN fails to adapt sufficiently while testing on new levels. This is consistent with [25], where $RL^2$ fails to adapt even after simplifying the level such that the coin is visible at the beginning of each trial.

# 4 RLBench Experiments

RLBench is a vision-based manipulation benchmark and learning environment, with a focus on sparse rewards, multi-task learning, and meta-learning. The environment has a rich set of more than 100 real-world inspired tasks involving diverse objects. The environment also provides easy access to expert demonstrations for all tasks, which is vital for overcoming the exploration problem imposed by the benchmark's sparse rewards. We classify RLBench as the medium difficulty benchmark in this study; while the embodiment is consistent across tasks (i.e., the Franka Panda robot arm), the task objects and goals vary drastically.

To ensure the experiment results do not get affected by arbitrary task selection, we design a

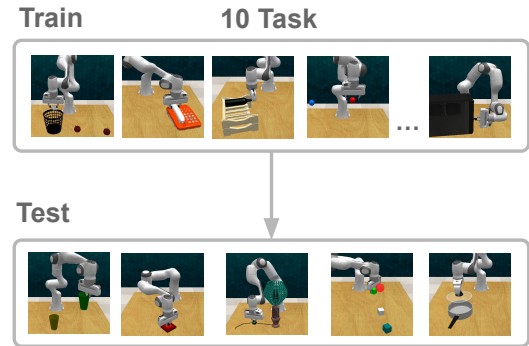

Figure 5: In RLBench experiments, each agent is trained on 10 tasks and evaluated on 5 unseen tasks.

comprehensive set of train-test task splits that resemble cross-validation in the supervised learning setting. Specifically, we use a fixed set of 11 RLBench tasks and create 5 splits. Each split uses a (randomly selected) held-out task and trains an agent on the remaining 10 tasks.

### 4.1 Training Setup

We use C2F-ARM [19] as the base off-policy RL algorithm. This was chosen because more widely-used RL algorithm, such as DDPG [28], TD3 [29], SAC [30], and DrQ [22] are known to fail [31] in RLBench due to the challenging setup. C2F-ARM [19] is a vision-based robot manipulation algorithm that can learn sparse-reward reinforcement learning tasks by using a small number of initial demonstrations. C2F-ARM is described in more detail in Appendix C. Note that RL$^2$ is excluded from this section because it is on-policy.

**Reptile-C2F-ARM** modifies the off-policy batch update in C2F-ARM with an inner- and outer-loop proposed in Reptile [11]. At the beginning of training, each task is given a separate replay buffer, which is initialized with transitions collected from 5 demonstration trajectories and continuously appended with the agent's online experiences. During training, for multiple steps in the inner loop, the agent draws a batch from the replay buffer of a randomly sampled task and performs updates to the Q-attention. In the outer loop, the network gets a soft update to mix the parameters from before and after the inner loop updates.

**PEARL-C2F-ARM** conditions a context embedding to the Q-attention network. To obtain the context for a task, a batch of transitions is drawn from a window of recent agent experiences, and a separate convolution encoder is used to first encode the image observations individually. Then, each image embedding is concatenated with the action and reward, and together encoded into a single vector. Finally, the context embeddings are sampled as proposed in [12]. The context encoder is additionally trained with a KL loss.

**MT-C2F-ARM** jointly trains C2F-ARM on all training tasks. During each replay batch update, both MT-C2F-ARM draw samples from multiple task replay buffers. During each replay update, a fixed number of tasks (less or equal to the total number of available training tasks) are randomly selected, then an equal number of samples are drawn for each task to construct the replay batch.

### 4.2 Test-time Adaptation Setup

Both MT-C2F-ARM and Reptile-C2F-ARM use the same C2F-ARM update and adapt the agent parameters to the new task via gradient descent. Adaptation for PEARL-C2F-ARM is done by gathering rollout samples in the new environment and re-computing the context embeddings, hence running only inference on the agent's policy model.

### 4.3 Results

The first set of evaluations are the most challenging for adaptation: an unseen **test-time task** given **0 demonstrations**. The agent is expected to leverage knowledge and skills gained in the 10 training tasks and perform intelligent exploration on the test task, without any guidance from demonstrations. Results for this setup are presented in the top row of Figure 6. Across all 5 test tasks, multi-task fine-tuning performs equally as well as Reptile while performing significantly better than both PEARL and training from scratch.

We next investigate the effect of reward sparsity on test-time performance. We now provide test-time demonstrations of each of the methods, as an aid for exploration under sparse reward. Results in the second and third row of Figure 6 show how the methods behave when given 1 and 2 *test-time demonstrations*. The fact that increasing the number of demonstrations improves training from scratch performance is unsurprising, however, one intriguing observation is that this effect is less apparent for MT-C2F-ARM and Reptile-C2F-ARM methods. This is encouraging evidence that **fine-tuning significantly reduces (or even omit) the need for demonstrations in sparse rewarded tasks, with little loss to performance.** We further investigate the various properties of fine-tuning C2F-ARM in Appendix C.

Apparent from Figure 6 is that PEARL does not seem equipped to handle such a disjoint train-test split. Recall that PEARL adapts without model parameter updates, and the only way to understand a new task is via aggregating new experiences into the context. However, the context encoder clearly fails at providing a useful context for the unseen tasks. we hypothesize this is due to our tasks setup: the training tasks are so visually disjoint that the agent never needs to learn high-quality context embeddings to infer which task it should do. This is different from the original experiment setup in PEARL, where *variations* are treated as "tasks", meaning that the observations from different "tasks" are similar or even identical; in order to disentangle the correct task to perform, the network

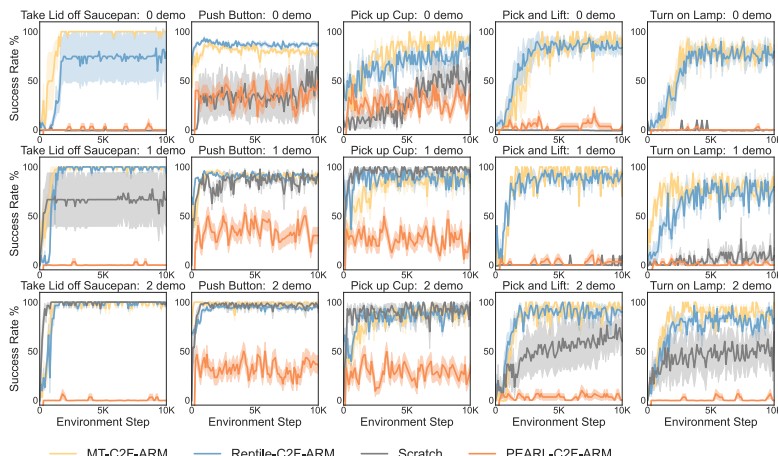

Figure 6: When varying the number of test-time demonstrations (from 0-2), does multi-task pretraining and fine-tuning outperform meta-RL methods on unseen tasks? We perform a "cross-validation" style evaluation: from a set of 11 RLBench tasks, 1 is held out for test-time evaluation, while the other 10 are used for training (and are given 5 demos). This is done for each of the 5 tasks above. Multi-task pretraining and fine-tuning perform equally as well or better than meta-RL. Unsurprisingly, fine-tuning (from either the Multi-task or Reptile agent) requires fewer demonstrations than training from scratch. Solid lines are average over 5 seeds, with shaded regions representing standard errors

is heavily motivated to read the context. Further evidence towards this hypothesis can be seen by looking at the zero-shot performance of the PEARL agent (i.e., environment steps = 0), where it starts with the same performance as multi-task and Reptile agents but doesn't improve. This suggests the meaningful performance that PEARL does achieve should be attributed to the pretraining and not test-time adaptation.

## 5 Atari Experiments

The Arcade Learning Environment (ALE, or the Atari benchmark) is one of the most commonly used environments within the RL community; however, rather than using the benchmark to train individual agents on each game, we use it to study meta-RL. We classify Atari as the hardest benchmark in this study due to the fact that the embodiment, objects, and goals, all vary across tasks; i.e., there is very little overlap between two different games.

Prior work has studied transfer between Atari games on relatively small scales, such as training an RL agent on **one** game and transfer to a **visually similar** game via fine-tuning [32, 33]. To the best of our knowledge, no prior meta-RL algorithm has been meta-trained on a subset of Atari games and tested on other games. Recently, *Oh et al.* [34] demonstrated meta-training via discovering RL update rules can transfer from a mini-grid environment to Atari, but performance still falls short from state-of-the-art results on training Atari games from scratch.

### 5.1 Training Setup

**Reptile-Rainbow** incorporates Reptile [11] with the off-policy batch update in Rainbow. Similar to Reptile-C2F-ARM, transitions from each Atari game are stored in a separate replay buffer. For multiple steps in the inner loop, the agent draws a batch from a randomly sampled buffer and performs updates to the distributional value network. In the outer loop, the value network gets a soft update to mix parameters before and after the inner loop updates.

**PEARL-Rainbow** conditions a context embedding to the value network. During each batch update, samples from each task are drawn from a recent window of agent experiences and get encoded as the context. The context encoder is additionally trained with a KL loss [12].

**MT-Rainbow** jointly trains RainbowDQN on all training games/tasks. In contrast to Reptile-Rainbow, the replay batch in both MT-Rainbow and PEARL-Rainbow contains samples from multiple task replay buffers. During each replay update, we randomly select a fixed number of tasks (less or equal to the total number of available training tasks), then draw an equal number of samples for each task and construct the replay batch.

Note that, Atari games generally have different action dimensions, hence for all pretrained agents, the action space is padded to be the maximum possible size (discrete, 18-dimensional). The extra dimensions are all mapped to "No-op" for the games with smaller original action space. For training from scratch, the agents are trained on the original action dimensions on their corresponding test game.

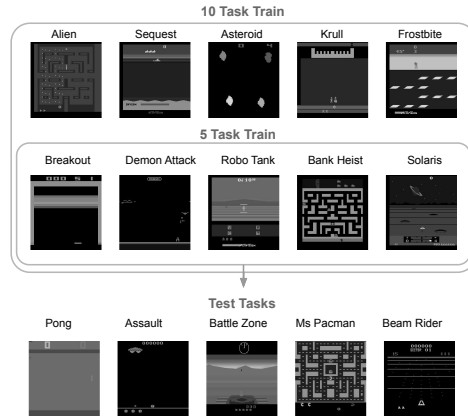

Figure 7: In Atari experiments, each agent is trained on up to 10 tasks and evaluated separately on 5 unseen tasks.

## 5.2 Test-time Adaptation Setup

We use 5 test-time tasks, namely Pong, Assault, Battle Zone, Ms Pacman, and Beam Rider. For Reptile-Rainbow and MT-Rainbow, a trained agent is finetuned with Rainbow for 100,000 environment steps on each unseen game, following the data-efficient benchmark proposed in [35]. For PEARL-Rainbow, the agent is rolled-out on a new game for 100,000 environment steps, and only updates the context embedding by encoding newly collected experiences.

We use the data-efficient version of Rainbow-DQN [20] as the base RL algorithm, i.e. we use the same training hyper-parameters and image encoder network architecture as proposed in [35]. Note that $RL^2$ is excluded from this section because it is on-policy.

For both Reptile-Rainbow and MT-Rainbow, all network parameters are finetuned. We provide additional results in Appendix D for loading only the convolutional encoder and re-initializing all MLP layers during finetuning. This setup was previously investigated in [33] at a smaller scale, i.e. training a Breakout agent and fine-tuning on Pong, but the network architecture and base RL algorithms differ from ours.

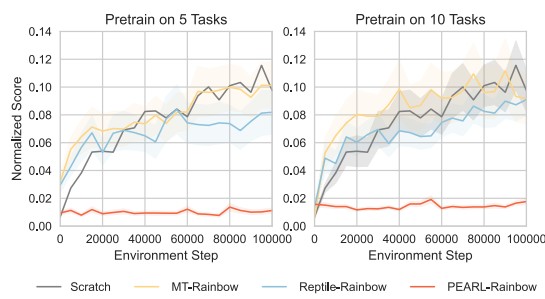

Figure 8: Adaptation on unseen Atari games. Solid lines are average over 10 seeds, with shaded regions representing standard errors.

## 5.3 Results

We report the test-time adaptation performance for each pretrained agent in Figure 8. The results are divided into 3 sets, based on the number of training tasks provided to each compared algorithm. For each set of experiments, the same training from "scratch" baseline is used (colored in gray).

For 5-task- and 10-task- trained agents, we observe high variance across different test-time games despite being finetuned from the same pretrained agent. For example, finetuning the Reptile-Rainbow agent trained on 10 tasks provides a clear advantage over training from scratch on Assault, but falls short on Ms Pacman and Beam rider.

Consistent with results on RLBench, PEARL-Rainbow also fails to adapt to the unseen task despite continuously updating its context embedding for a sufficiently long period. PEARL-Rainbow's zero-shot performance on some of the games is clearly better than random (i.e. the first data point for training from scratch)— however, this benefit comes more from the diversity of training tasks than from the meta-training algorithm itself, which is evident from observing that: **(1)** PEARL-Rainbow has similar zero-shot performance to that of MT-Rainbow and Reptile-Rainbow, and **(2)** as the number of training task goes from 5 to 10, PEARL-Rainbow's performance slightly improves. Overall, training from scratch has competitive performance across all test-time tasks, which is in contrast to previous results on Procgen and RLBench. Previous work has also found that transfer between Atari games is challenging [36, 32, 37], and the common hypothesis is that the visuals and control strategies vary too much across games for positive transfer. We provide further empirical results in Appendix D to gain more insights into this hypothesis.

## 6 Summary of Compared Algorithms

In order to allow easy side-by-side comparison of how different methods are implemented in our experiments, we provide a unified overview of the algorithms that are as shown in Algorithms 1 and 2.

---

**Algorithm 1** Training On-policy Meta-RL

**Require:** Learning rate $\alpha$; Reptile step size $\epsilon$;
Policy optimization objective $\mathcal{L}$; Trajectory buffer $\mathcal{D}$
**Input:** Algorithm name $algo$
Initialize policy $\pi_\theta$, LSTM embedder $\phi$
**while** not converged **do**
  **if** $algo ==$ Reptile **then**

    Sample a single task $\mathcal{T} \sim p(\mathcal{T})$
    Set $\theta^{old} \leftarrow \theta$
    **for** $i = 0$ **to** $k - 1$ **do**
      Collect trajectories $\mathcal{D}^i$ using $\pi_{\theta^i}$
      $\theta^{i+1} \leftarrow \theta^i - \alpha \nabla_{\theta^i} \mathcal{L}(\pi_{\theta^i}, \mathcal{D}^i)$
    **end for**
    $\theta \leftarrow \epsilon(\theta^{old} - \theta^k)$

  **else if** $algo ==$ RL$^2$ **then**

    **for** $i = 1$ **to** $b$ **do**
      Sample task $\mathcal{T}_i \sim p(\mathcal{T})$; collect $\mathcal{D}^i$ using $\pi_{\theta^i}, \phi$
    **end for**
    $\theta \leftarrow \theta^i - \alpha \nabla_\theta \sum_{i=1}^b \mathcal{L}^i(\pi_\theta, \phi, \mathcal{D}^i)$
    $\phi \leftarrow \theta^i - \alpha \nabla_\phi \sum_{i=1}^b \mathcal{L}^i(\pi_\theta, \phi, \mathcal{D}^i)$

  **else if** $algo ==$ Pretrain **then**

    **for** $i = 1$ **to** $b$ **do**
      Sample task $\mathcal{T}_i \sim p(\mathcal{T})$; collect $\mathcal{D}^i$ using $\pi_\theta$
    **end for**
    $\theta \leftarrow \theta - \alpha \nabla_\theta \sum_{i=1}^b \mathcal{L}(\pi_\theta, \mathcal{D})$

  **end if**
**end while**

---

**Algorithm 2** Training Off-policy Meta-RL

**Require:** Learning rate $\alpha$; Reptile step size $\epsilon$
Context sampling heuristics $\mathcal{S}_c$; Value-function loss $J_Q$;
KL loss $\mathcal{L}_{KL}$;
**Input:** algorithm name $algo$
Initialize Q-function $Q_\theta$, context encoder $\phi$;
**while** not converged **do**
  Sample $B$ replay buffers and draw samples from each buffer to construct batch $b = \{b^1, ..., b^j\}$
  **if** $s ==$ Reptile **then**

    **for** $j = 1$ **to** $B$ **do**
      $\theta_j^{old} \leftarrow \theta$
      **for** $i = 0$ **to** $k - 1$ **do**
        $\theta^{i+1} \leftarrow \theta^i - \alpha \nabla_{\theta^i} J_Q(b^j)$
      **end for**
      $\theta \leftarrow \epsilon(\theta^{old} - \theta_j^k)$
    **end for**

  **else if** $algo ==$ PEARL **then**

    **for** $j = 1$ **to** $B$ **do**
      Sample contexts $c^j = \mathcal{S}_c(\mathcal{D}_i)$
      $\mathcal{L}_{KL}^j = \mathcal{L}_{KL}(\phi(c^j))$
    **end for**
    $\phi \leftarrow \alpha \nabla_\phi \sum \mathcal{L}_{KL}^j$
    $\theta \leftarrow \theta - \alpha \nabla_\theta J_Q(b, \{c^1, ..., c^j\})$

  **else if** $s ==$ Pretrain **then**

    $\theta \leftarrow \theta - \alpha \nabla_\theta J_Q(b)$

  **end if**
**end while**

---

## 7 Related Work

**Meta-Reinforcement Learning**  Meta-RL aims to find the best learning strategy that enables fast adaptation to a new task via reinforcement learning. This often relies on meta-training with a distribution of tasks and exploiting their shared structures. Two main approaches include context-based methods and gradient-based methods. Context-based methods are trained to use recent rollout experiences from a new task to form a context that can be used to distinguish what task the policy is solving. Previously, this context has been formed implicitly via an LSTM [16, 38], or explicitly, by passing trajectories through a separate encoder, whose output is given to a context-conditioned policy [12, 39]. Gradient-based methods perform test-time optimisation of hyperparameters [40], loss functions [41, 42], or network parameters [3, 43, 44].

The meta-RL approaches above have only been studied in fully observable state settings with shaped rewards; neglecting more realistic real-world scenarios, where rewards are often sparse, and observations are high-dimensional (e.g. images, point clouds, etc). There is limited work that study

these issues: for example, hindsight relabeling is used to aid in sparse reward setups e.g. [45], but uses fully observable states. Other approaches to sparse reward and partial observability include HyperX [46], DREAM [47], and MetaCure [48]. Out-of-distribution variation adaptation within the same task is another challenging setup, where recent methods include model-identification, experience relabeling, [49], and adding symmetries [50].

Beyond context-based and gradient-based methods built on model-free RL algorithms, other lines of work include: **model-based meta-RL**, via meta-training a dynamics model and has seen success in enabling adapting to different hardware or terrain conditions on a legged millirobot [51]; **meta-imitation learning**, has been applied to vision-based robot manipulation [10, 52, 53]; **meta-learn RL algorithms**, which aims to discover RL objectives or update rules that can be transferred across different task environments [34, 54–57].

Although in our experiments, we follow the original designs pf PEARL and RL$^2$ which don't allow gradient updates during test time, recent work [58] has looked into the theoretical limitations of context-based meta-RL algorithms in out-of-distribution variation adaptation setting [58], and shown that adding gradient updates (i.e. finetuning) at test time helps improve adaptation.

**Multi-task Reinforcement Learning**   The pretraining procedure in our experiments is multi-task reinforcement learning (MTRL), where the training objective is simply finding a single best policy across multiple tasks. The main challenge in multi-task learning in general lies in multi-objective optimization, and has been investigated in MTRL [59, 60] and applied to robotics [61]. Recently, *Kurin et al.* [62] demonstrated that joint training with proper regularization achieves competitive performance with the more complicated multi-task algorithms. This observation aligns with our multi-task training results.

**Meta- v.s. Multi-task pretraining in RL**   Multi-variation pretraining followed by fine-tuning, also called domain random search (DRS), is also shown to achieve comparable performance to meta-RL on existing state-based benchmarks [63]. Our work expands on this setup by training on more distinct **tasks** instead of variations, and excluding the test-time task from training. Notably, the meta-learning suite (e.g. ML10, ML45) in the MetaWorld [64] benchmark also poses such task generalization challenges, and finetuning is recently shown to be better than meta-RL algorithms such as RL$^2$ and MAML [24].

# 8   Conclusion

We perform a large-scale study on vision-based meta-RL across a *truly* diverse set of tasks. Our results show that when trained and tested on truly diverse reinforcement learning tasks, simple pretraining and fine-tuning can perform equally as well as well as more complex meta-learning approaches. This is consistent with the findings within the computer vision community [4–7], but in contrast to the large body of current literature, which shows that meta-RL is effective when evaluated on *variations* of the same train-time task.

Our work is an initial step towards understanding the subtleties between meta-RL and multi-task pretraining, with plenty of room for further investigation. Despite the breadth of our experiments across 3 benchmarks, a limitation of our study is the small number of available pre-training tasks for both RLBench and Atari. From the experimental results, we remark that training on such limited number of training tasks may not be sufficient to fully learn the representations required to enable the best adaptation performance on new tasks. However, note that the purpose of this study was not to show that multi-task pretraining followed by fine-tuning is better than training from scratch, but rather that multi-task pretraining can perform equally as well, or better, than meta-RL.

## Acknowledgments and Disclosure of Funding

This work was supported by DARPA RACER and Hong Kong Centre for Logistics Robotics. The authors would like to thank Shikun Liu, Danijar Hafnar, Olivia Watkins, Yuqing Du, and Luisa Zintgraf for their valuable discussions and helpful feedback on initial drafts of the paper.

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
