# A Detailed justifications for the checklist questions

1(a) Do the main claims made in the abstract and introduction accurately reflect the paper's contributions and scope?[Yes] We believe our experiments provide an extensive study to our problem statement, comparing multi-task pretraining and finetuning with meta-RL. The empirical results are comprehensive and clearly supports our stated contributions.

(b) Did you describe the limitations of your work? [Yes] Yes, see section 8

(c) Did you discuss any potential negative societal impacts of your work?[Yes] Yes, see section 8

(d) Have you read the ethics review guidelines and ensured that your paper conforms to them?[Yes]

2. If you are including theoretical results...[N/A]

3. If you ran experiments...

(a) Did you include the code, data, and instructions needed to reproduce the main experimental results (either in the supplemental material or as a URL)? [Yes] Please refer to the code scripts provided in the supplemental material.

(b) Did you specify all the training details (e.g., data splits, hyperparameters, how they were chosen)? [Yes] Please refer to both main text and appendix for experiment details.

(c) Did you report error bars (e.g., with respect to the random seed after running experiments multiple times)?[Yes]
All adaptation experiments in Procgen and RLBench are run for 3 seeds. In Atarim all adaptation experiments are run for 10 seeds due to the high variance during RL training.

(d) Did you include the total amount of compute and the type of resources used (e.g., type of GPUs, internal cluster, or cloud provider)? [Yes]
As stated in section 2, we use RTX A5000 GPUs each with 24GB memory. Procgen experiments are all done on single-GPU: it took 10 hours for training each multi-task agent for 100M environment steps, and 10 minutes for each finetuning/adaptation run on each of the 20 test levels. RLBench experiments use 4 GPUs for each run; training each multi-task agent took 24 hours, and finetuning/adaptation on each test task for each of the 3 seeds takes 9 hours. Atari experiments are done on single GPUs; training each agent takes 24hrs for 1 million steps, and finetuning/adaptation on each test game took 1 hours for each of the 10 seeds.

4. If you are using existing assets (e.g., code, data, models) or curating/releasing new assets...

(a) If your work uses existing assets, did you cite the creators? [Yes]
We use the Procgen RL environment released by OpenAI under MIT lisence [13]; the RL training code is built on the open-source RL framework implementation from Stable-baselines3 [65]
We use the RLBench simulated robotic manipulation environment [8] released under MIT license. Our C2F-ARM algorithm and training framework are built based on the original author's implementation and open-sourced code under MIT license.
We use The Arcade Learning Environment [15, 14] (under MIT license) for simulated RL environments in our Atari experiments. The code for RainbowDQN is built on the open-source implementation [66]

(b) Did you mention the license of the assets? [Yes] See described above, all used assets are released under MIT license.

(c) Did you include any new assets either in the supplemental material or as a URL? [Yes] Yes, please see the supplemental material for all the code for reproducing our experiments.

(d) Did you discuss whether and how consent was obtained from people whose data you're using/curating? [N/A]

(e) Did you discuss whether the data you are using/curating contains personally identifiable information or offensive content? [N/A]

5. If you used crowdsourcing or conducted research with human subjects... [N/A]

# B Additional Procgen Remarks

## B.1 Experiment Details

**Hyperparameters** See Table 1 for hyperparameter settings used for RL training on Procgen. The base PPO training parameters are shared across all compared algorithms. Our code is built off the PPO training code implemented in Stable-baselines 3 [65].

**Task settings** See Figure 9 for additional visualizations of the training levels. We show 10 levels from each training set, note that each smaller training set is a subset of levels from the bigger training sets. All 20 test levels are shown under "test levels". Note the high diversity of color themes and layouts across different levels.

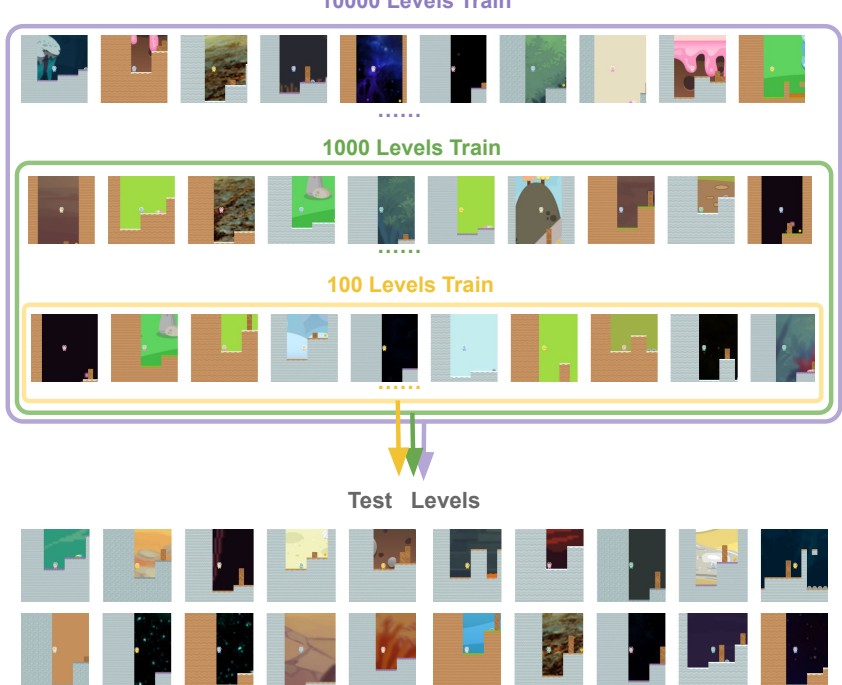

Figure 9: Visualization of all 20 test levels and a subset of training levels from game Coinrun in Procgen.

## B.2 Training Results for Procgen Experiments

The results on training levels for all compared methods are reported in Table B.2, where each column corresponds to training each agent with a different size of training set, i.e. the number of training-levels used. Each cell reports the final reward for a single agent that was trained for 100M environment steps, averaged over all training levels.

| Method | Average Reward on 100 levels | Average Reward on 1000 levels | Average Reward on 10000 levels |
|---|---|---|---|
| MT-PPO | 9.7 | 8.0 | 7.5 |
| $RL^2$-PPO | 4.7 | 7.7 | 7.8 |
| Reptile-PPO | 9.5 | 5.8 | 5.3 |

## C  Additional RLBench Remarks

### C.1  Coarse-to-fine Attention Driven Robotic Manipulation (C2F-ARM)

A core component of C2F-ARM is the coarse-to-fine Q-attention [19] module, which takes as input a coarse 3D voxelization of the scene, and learns to attend to interesting areas within the coarse voxelization. The module then 'zooms' into that area and re-voxelizes the scene at a higher spatial resolution. This 'attend-and-zoom' procedure is applied iteratively and results in a near-lossless discretization of the 3D space. This 3D space discretization is combined with a rotation discretization to give a continuous next-best pose output. C2F-ARM takes this next-best pose and uses a motion planner to take the robot to the goal pose. In this work, we use the original C2F-ARM algorithm, and do not include any of its subsequent extensions, e.g., learned path ranking [67] and tree expansion [68].

As mentioned above, a small number of demonstrations are used to overcome the exploration problem within these sparse-reward environments. The predecessor to C2F-ARM, ARM [31], introduced two demonstration pre-processing procedures: (1) *keyframe discovery*, which assists the Q-attention at the initial phase of training by suggesting meaningful points of interest; and (2) *demo augmentation*, which takes demo episodes and produces many sub-episodes with different starting points, thereby increasing the initial number of demos in the replay buffer. All C2F-ARM results use both pre-processing procedures.

Table 1: Hyperparameters for Procgen Experiments

| PPO | Value |
|---|---|
| Environment steps (pretraining) | 100,000,000 |
| Environment steps (testing) | 2,000,000 |
| Mini batch size | 2048 |
| Learning rate | 5e-4 |
| Number of epochs | 3 |
| Discount factor $\gamma$ | 0.99 |
| GAE coefficient $\lambda$ | 0.95 |
| Clip Range | Constant 0.2 |
| Entropy coefficient | 0.01 |
| Value function coefficient | 0.5 |
| Gradient clipping norm | 0.5 |
| Target KL divergence | 0.01 |

| Reptile-PPO | Value |
|---|---|
| Number of inner-loop iterations | 3 |
| Soft parameter update schedule $\epsilon$ | Linear 0-1 |

| $RL^2$-PPO | Value |
|---|---|
| LSTM hidden state dimension | 256 |
| LSTM number of hidden layers | 2 |

## C.2 Training Results for RLBench Experiments

We report results on training tasks for all compared methods in Table C.2, where each row corresponds to training with one task held-out and using only the remaining 10 tasks. Each cell reports the final success rate for a single agent on all training tasks, each evaluated 10 episodes and results are averaged across tasks.

| Held-out Task | MT-C2FARM Avg. Success Rate | PEARL-C2FARM Avg. Success Rate | Reptile-C2FARM Avg. Success Rate |
|---|---|---|---|
| Lid off Saucepan | $64.64 \pm 9.89$ | $49.88 \pm 11.45$ | $71.91 \pm 10.58$ |
| Push Button | $60.93 \pm 12.29$ | $62.86 \pm 9.73$ | $55.97 \pm 13.58$ |
| Pick and Lift | $65.30 \pm 10.15$ | $51.19 \pm 12.99$ | $68.84 \pm 12.67$ |
| Pick up Cup | $67.30 \pm 9.40$ | $53.17 \pm 11.00$ | $47.74 \pm 11.70$ |
| Turn on Lamp | $68.63 \pm 10.78$ | $57.80 \pm 11.21$ | $54.97 \pm 12.85$ |

## C.3 Single-task, Multi-variation Experiments

As a sanity check for whether the meta-RL algorithms are able to generalize to an easier, multi-variation setup, we experiment with the multi-variation *push_button* task from RLBench, where the task is to push a button on the tabletop but different variations differ in button colors. Each method is trained on 10 variations and average test-time performance on 5 unseen variations. Results are reported in 10: all compared methods can perform an unseen variation in a zero-shot manner, and PEARL is able to adapt despite the lack of gradient updates.

## C.4 Properties of fine-tuning Q-attention

Earlier in the paper, we have shown that multi-task pre-training, followed by fine-tuning, can perform equally as well as meta-RL. In the following set of experiments, we explore fine-tuning in more detail, focusing on: **(1)** zero-shot performance on test tasks (no test-time gradient updates); **(2)** investigate whether it is better to fine-tune an unseen task in isolation, or together with other tasks (in a multi-task setup); and **(3)** inspecting the role of each Q-attention depth on fine-tuning performance.

We begin by evaluating zero-shot task performance on held-out test tasks when pretrained with multi-task pretraining and evaluated on 30 episode rollouts. Results in Table 2 show that multi-task pretraining, even on a small number of tasks, can allow the Q-attention to begin to learn an intuition of 'objectness', which can be useful for zero-shot performance on some tasks.

The next set of experiments aims to investigate whether it is better to fine-tune an unseen task in isolation, or together with other tasks (in a multi-task setup). The intuition for the former is that train-time tasks (where we have access to demos), can be used to learn good representations and exploration strategies; while the latter intuition is that mixing with train-task data can act as auxiliary tasks, and the test-time task is treated as the main

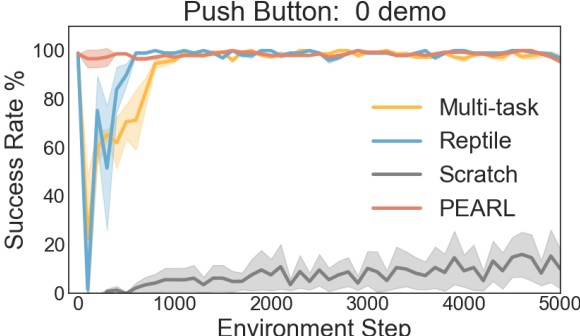

Figure 10: Adaptation result on unseen variation of the "push button" task on RLBench. Each method is trained on 10 variations and average test-time performance on 5 unseen variations. All compared methods can perform an unseen variation in a zero-shot manner, and PEARL is able to adapt despite the lack of gradient updates.

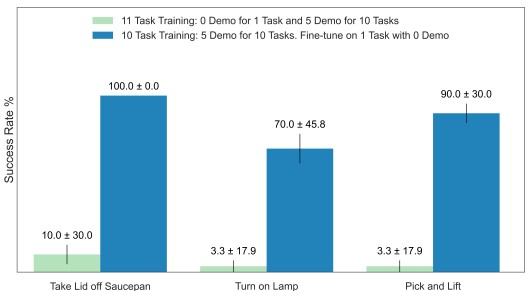

Figure 11: Is it is better to fine-tune an unseen task in isolation, or together with other tasks? We perform a "cross-validation" style evaluation, where from a set of 11 RLBench tasks, 1 is held out for test-time evaluation (and given 0 demos), while the other 10 are used for pretraining (and are given 5 demos). This is done for each of the 3 tasks above. Each color bar represents the average evaluation over 30 episodes while the error bars represent the standard deviations.

task. As shown in Figure 11, fine-tuning in isolation is superior to training in a multi-task setting. The hypothesis here is that the agent can keep the representations and skills that are useful to the fine-tune task, while forgetting non-useful ones; whereas training with other tasks requires that the network have the capacity to remember all skills.

The final set of experiments aim to inspect the role of each Q-attention depth on fine-tuning performance. Figure 12 shows a comparison of three different fine-tuning strategies: (**1**) updating only the first (coarse) Q-attention depth, (**2**) updating only the second (fine) depth, and (**3**) updating both depths. Unsurprisingly, updating both depths gives the best performance, however, fine-tuning only the second depth (while leaving the first depth fixed) performs almost equally as well. This suggests that during pretraining, the Q-attention learns a good understanding of 'objectness' at the coarse level (i.e., *what* object to interact with), while the "fine" level is more concerned with *how* to interact with the object, which is more task-specific, and therefore has the most benefit from fine-tuning.

Table 2: Zero-shot task performance on held-out test tasks, when pretrained with multi-task pretraining. In column "Success Rate (Train)", we report the final training performance averaged across evaluating 30 episodes for each of the 10 training tasks. In column "Success Rate (Unseen Task)", we report zero-shot direct evaluation performance of the trained agent on the held-out unseen task.

| Held-out Task | Success Rate (Train) | Success Rate (Unseen Task) |
|---|---|---|
| Lid off Saucepan | $64.64 \pm 9.89$ | $0 \pm 0$ |
| Push Button | $60.93 \pm 12.29$ | $63.33 \pm 8.80$ |
| Pick and Lift | $65.30 \pm 10.15$ | $0 \pm 0$ |
| Pick up Cup | $67.30 \pm 9.40$ | $36.43 \pm 8.83$ |
| Turn on Lamp | $68.63 \pm 10.78$ | $3.33 \pm 3.28$ |

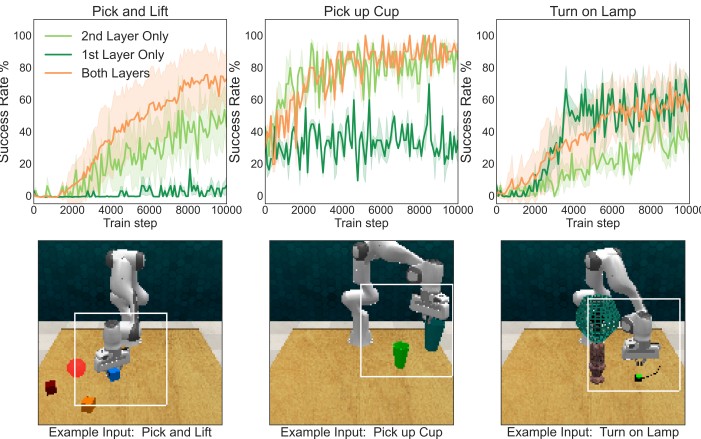

Figure 12: A deeper look at the role of Q-attention when fine-tuning on 3 unseen tasks. **Top:** we compare how performance for 3 different fine-tuning strategies: (1) updating only the first Q-attention depth, (2) updating only the second depth, and (3) updating both depths. **Bottom:** visualization of the scene information captured by the Q-attention; whole image represents Q-attention depth 0 which captures global, whole-scene information, while white square represents Q-attention depth 1 which captures local, fine-grained information.

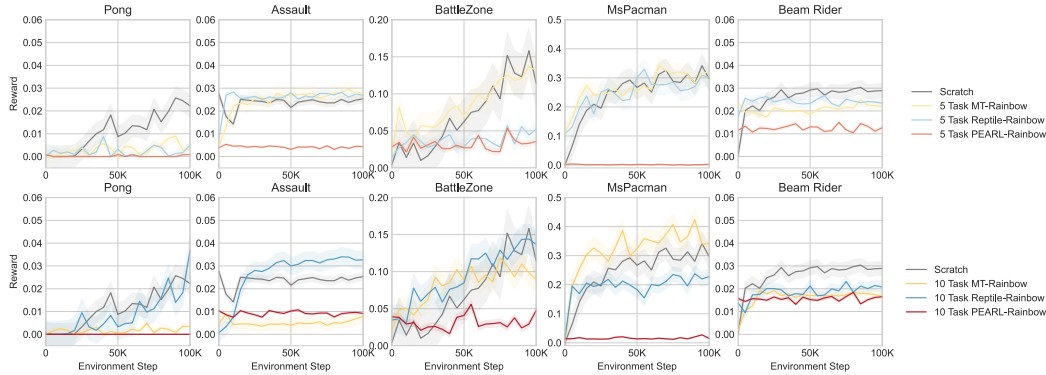

Figure 13: Adaptation results on all 5 test-time Atari games using two different training setups: meta-training or pretraining using 5 tasks (top row) and 10 tasks (bottom row). Each game's scores are normalized by the final converged result for each corresponding game as reported in RainbowDQN [20]. We plot the normalized scores averaged over 10 seeds for each method.

# D   Additional Atari Remarks

## D.1   Full adaptation results

See Figure 13 for separate adaptation results on all 5 test games. For each training method (either training from scratch, adaptation or finetuning), we run 10 seeds on every test game. Because we use the data-efficient benchmark [35] which does not train RainbowDQN to full coverage, we normalize the scores on each game by the corresponding final RainbowDQN result, as reported in [20]. We then plot the normalized reward averaged over all 10 seeds. The first and second row shows adaptation performance from two different training sets: meta-training or pretraining using 5 tasks (top row) versus 10 tasks (bottom row).

Hyperparameters for training base RainbowDQN (data-efficient version) are shared across all compared algorithms, see Table 3 for details. Our code is built on the popular open-source implementation from [66].

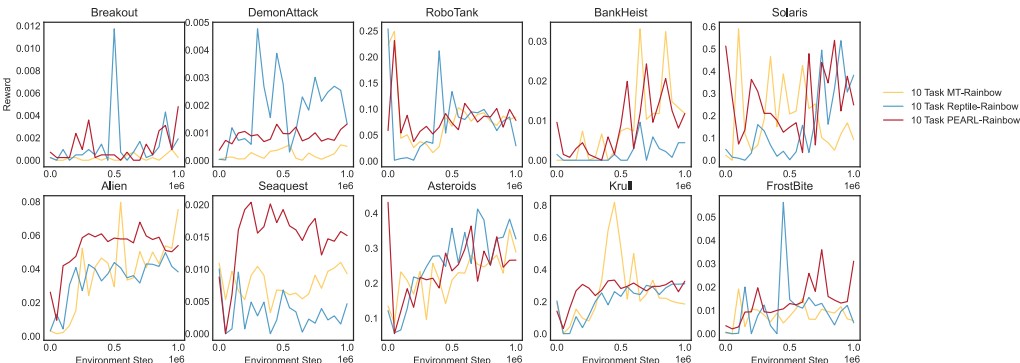

Figure 14: Training performance of the compared method on Atari. Results are reported separately on each training game.

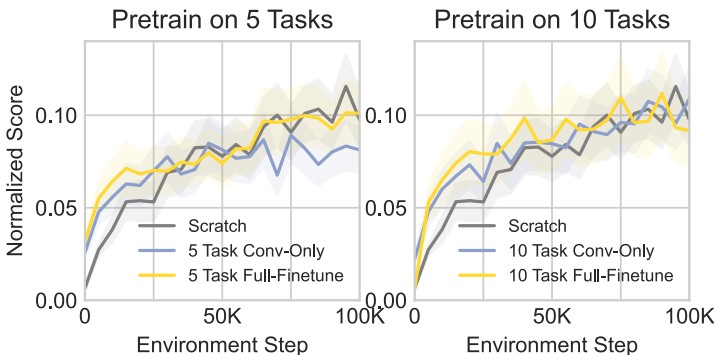

Figure 15: Finetuning the entire network versus finetuning with a partially-loaded (only convolutional layers) agent. Averaged performance across all 5 test-time Atari games shows little difference between the two settings, and shows similar trend when pretrained with different number of tasks (5 v.s. 10). When additionally compared with training from scratch, the results suggest that the transferability between Atari games is likely to be too low for pretraining

## D.2   Training Results for Atari Experiments

We report the performance of the three compared method during training in 14 : each method is trained on the same set of 10 Atari games, and rewards are normalized in the same way as the adaptation results, i.e. normalized by the per-game results reported in RainbowDQN [20] (since our setup follows the data-efficient setup in [35], the results are lower than reported in [20]). Notice that, the overall performance of all three methods (MT-Rainbow, Reptile-Rainbow, and PEARL-Rainbow) are similar across each training game, which suggests that their corresponding adaptation performances are not particularly held-back or aided by the performance on training tasks.

## D.3   Additinal Finetuning Experiments

We compare two ways of finetuning a multi-task agent: finetune all network parameters, versus finetune only the convolutional layers and re-initialize the MLP layers. Results are reported in Figure 16 We note again the small difference between the two sets of results and between finetuning and training from scratch, which further suggests the possibility that there is little shared knowledge between Atari games that can be transferred.

## D.4   Finetuning PEARL-Rainbow Experiments

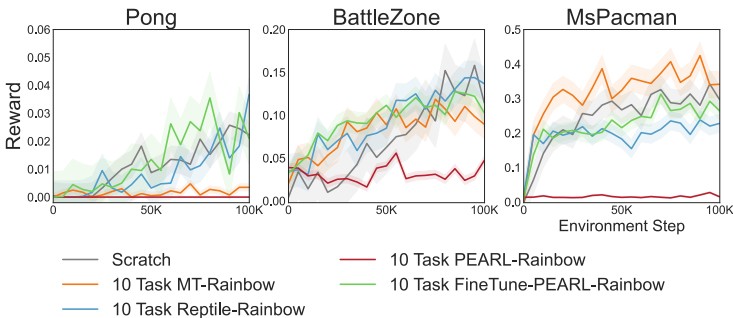

Figure 16: Can PEARL adapt to the new unseen games via finetuning? We finetune a PEARL agent, which was pretrained on 10 Atari games, on 3 separate test-time games. In contrary to gradient-free adaptation (shown in red), finetuning the PEARL-Rainbow agent (shown in green) allows it to adapt to the unseen games, and achieves similar performance to finetuning the multi-task agent and the Reptile agent.

Table 3: Hyperparameters for Atari Experiments

| RainbowDQN | Value |
| --- | --- |
| Environment steps (pretraining) | 1,000,000 |
| Environment steps (testing) | 100,000 |
| Batch size | 32 |
| Learning rate | 1e-3 |
| Minimum of value distribution support | -10 |
| Maximum of value distribution support | 10 |
| Multi-step return | 20 |
| MLP hidden dimension | 256 |
| Action dimension | 18 |
| **Reptile-Rainbow** | **Value** |
| Number of inner-loop updates | 5 |
| Soft parameter update schedule $\epsilon$ | Linear 0-1 |
| **PEARL-Rainbow** | **Value** |
| Context embedding dimension | 32 |
| KL loss coefficient | 1 |