# OpenReview forum: "On the Effectiveness of Fine-tuning Versus Meta-reinforcement Learning "
_NeurIPS.cc/2022/Conference — NeurIPS 2022 Accept_

### Official Review · Reviewer_DrBc · 2022-07-10

**Rating:** 4
**Confidence:** 4
**Soundness:** 3 good
**Presentation:** 3 good
**Contribution:** 2 fair

**Summary:**

The authors recap the pretrain-finetune framework for the generalisation RL problem and compare this simple baseline with many Meta-RL algorithms. They conduct the comparision on three vision-based RL tasks: Procgen, RLBench, and Atari. The final results show that in most case multi-task pretraining with fine-tuning can achieve comparable or even better performance than Meta-RL algorithms.

**Questions:**

(1) As far as I know, most Meta-RL algorithms are based on policy-based algorithms rather than Q-value based algorithms. In the experiments, 4 meta-rl algorithms are using Q-value based algorithms: Reptile-C2F-ARM, PEARL-C2F-ARM, Reptile-Rainbow, PEARL-Rainbow. For Reptile-C2F-ARM and Reptile-Rainbow. they are gradient-based Meta-RL and are meta-trained based on TD-error. However, in original MAML the meta-rl agent is trained with policy gradient. Though direct extension from policy-based MAML to value-based MAML seems to be technically sound, no further paper make it work empirically. For PEARL-C2F-ARM and PEARL-Rainbow, I also doubt whether there exist some papers make context-based value-based RL algorithm work on traditional meta-rl environments. Thus, I believe some meta-rl algorithms used in this paper are inappropriate.

(2) Can you give more explainations on why you choose different task settings (described in Assumption on Task Distribution). If the state/action space vary a lot between different task, it more relies on the generalisation of observation space (as the author mentioned: "An agent should be able to infer which task to perform based on a single-step observation" rather than the generalisation of MDP. I think this is also the reason why RL generalisation and Meta-RL are sperate research branches because they are focusing on different problem.

(3) It is not surprising to see that context-based Meta-RL algorithms such as Pearl and RL2 cannot work well for most experiments since the adaption steps are very large like 10k/1M. I believe on small RNN or simple MLP network cannot handle such large history information (especially your RNN hidden size is only 256). Have you tried more comlex network structure or more parameters to see if it can help increase the performance?

(4) For vision-based meta-rl task, why not compare pretrain-finetune to the vision-bsed meta-rl algorithms like MELD [1]?

(5) Can you also present the performance of the pretraining process? If the algorithms cannot be pretrained on the training distribution well, it is not surprising to see it cannot generalise on test distribution.


[1] Zhao T Z, Nagabandi A, Rakelly K, et al. Meld: Meta-reinforcement learning from images via latent state models[J]. arXiv preprint arXiv:2010.13957, 2020.

**Limitations:**

Overall, this paper presents an interesting idea to reconsider the benchmark/evaluation protocol for Meta-RL. However, I think the main task setting and some baseline algorithms are inappropriate, which makes the final conclusion weak. I highly recommend the authors to try to start from some traditional meta-rl benchmarks. Because they tend to have well-tuned meta-rl baselines and can make the final comparision more fair. For instance, there also exists some complicated meta-rl benchmark such as Meta-World, where the task variation happens beyond reward/dynamic transitions. Some further experiments can be done there to strenghen your argument.

**Strengths And Weaknesses:**

Strengths:

(1) Overall this paper offers a new view to consider a more realistic and reasonable evaluation protocol/benchmark for Meta-RL research.

(2) Thorough experiments are conducted over different RL benchmarks comparing many different algorithms/baselines.

(3) The writing is overall clear and easy to follow.

Weaknesses:

(1) Lack of more ablation studies analysing the reason why Meta-RL algorithms fail. (For instance, maybe because of the vision-based task's difficulty/it requires some many samples to adapt/lack of capacity of context network?)

(2) Inappropriate meta-rl baseline extensions. (See Questions 1)

---

> ### Author Response · Authors · 2022-08-02
> **Author response to reviewer DrBc - Pt1**
>
> Thank you for your feedback and suggestions for improvement. We hope the following could address the raised concerns and questions.
>
> 1) Weakness due to lack of more ablation studies analyzing the reason why Meta-RL algorithms fail.
>
> We’d like to clarify that we showed ‘failure’ only on context-based meta-RL, which requires no parameter update at adaptation, and gradient-based algorithm, i.e. Reptile, is still able to adapt to the new task with better performance than training from scratch. Hence our main focus on how fine-tuning performs just as well as Reptile, but more favorable due to its simplicity.
>
> Throughout the submission, we attributed this failure of PEARL and RL2 to the more challenging train & test task setup. In the updated appendix, we also added a new set of experiments (Figure 11) that uses a ‘easier’, multi-variation setup. The task used is “push-button” from the RLBench task suite, where each variation corresponds to a different button color that the robot arm should push. Results show that PEARL can perform the unseen variation at test-time, further suggesting that the training task distribution, although being a finite discrete set of different colors, sufficiently covers the test-time color and allows generalization.
>
> The vision-based task's difficulty and/or the algorithm’s sample efficiency does not seem to be the issue here, since we rolled out PEARL and RL^2 for a sufficiently long period of environment steps and did not see any sign of adapting. The network capacity also may not be the reason, because the training performance showed that the capacity is sufficient for fitting to the much larger training set of tasks. It’s indeed desirable to further understand why meta-RL doesn’t perform as expected, and we’d love to hear back if the reviewer has follow-up suggestions on how to concretely design new investigations.
>
>
> 2) Weakness due to meta-RL baselines are inappropriate.
>
> Contrary to the reviewer’s impression, there actually are many meta-RL algorithms that are built on off-policy RL, such as Meta-Q-Learning [1], MetaCure[2], and PEARL[3] (we updated the related work section to better include these references).  PEARL was originally proposed to be combined with off-policy RL algorithms, and implemented on top of Soft Actor Critic (see their openly available implementation here: https://github.com/katerakelly/oyster) — hence we believe it’s appropriate to combine it with C2F-ARM and RainbowDQN.
>
>
> 3) Question on whether RL2 and PEARL fail because the adaptation steps are too large and network capacity is too small:
>
> A gentle reminder that the actual episode length for the tasks we used are reasonably short: maximally 10 timesteps for RLBench tasks, and ~100 steps for Procgen. The Atari games are known to have varying episode length, e.g. Pong has ~20 steps and BattleZone has up to 10000 steps, but our results are consistent across these games. Although our figures show large step sizes for the x-axis (10k/1M), it only means plotting the agent performances at these intervals, and to show their performance does not improve after being rolled for very long periods. episode-length for tasks in RLBench and Procgen are short. In addition, the training performance suggests that the network capacity has no problem fitting to multiple training tasks, so it does not seem to be the bottleneck when we test by adapting on a single task at a time. In our implementation of PEARL and RL2, the context encoder / hidden LSTM can be seen as an additional network, and the rest of the agent network architecture is kept the same with the Reptile and Multi-task agents on C2FARM and RainbowDQN — hence there are actually more network capacity for PEARL and RL^2, but they still perform worse than fine-tuning.
>
> 4) Request for adding MELD as another baseline.
>
> In our understanding, MELD is a less widely adopted follow-up that’s most closely related to PEARL, hence we think it’s acceptable to only use PEARL. Notably, MELD is designed for vision-based, multi-_variation_ setup, and not our vision-based, multi-task setup. Based on their results (Figure 5), it’s indeed better at PEARL at adapting to new variations from image inputs, but it’s still unlikely to adapt better than PEARL when given completely unseen tasks.
>
> Similar requests can also be made for MQL [1], MetaCure[2], etc. Given our 3 benchmark setup, it’s quite resource heavy to add many meta-RL baselines to all the tasks, and we hence chose to only show the most representative ones. Nevertheless, we would love to follow up with more experiment results if the reviewer deems important that this baseline be included.

---

> ### Author Response · Authors · 2022-08-02
> **Author response to reviewer DrBc - Pt2**
>
>
> 5. Request for results on the performance of the pretraining process.
>
> Please see the updated appendix for pretraining performance. We remark that on all task setups, the compared methods (meta-RL and fine-tuning) yield similar performances. This suggests that the algorithms are well prepared to acquire the skills needed for training, but only the adaptation part is not any better than the simpler approach of directly fine-tuning the model parameters.
>
>
> 6. Request for shifting to using “traditional” meta-RL benchmarks:
>
> We emphasize that there already exists prior work that investigates meta-RL on state-based environments, and its comparison to fine-tuning  [3,4]. Studying vision-based environments is a valuable setup that can be well placed into the literature, and we hope the reviewer would not perceive it as a limitation.  A less ‘traditional’ benchmark is Metaworld [5], which provides multi-task with state-input settings, but MAML, RL2, PEARL are shown to give poor adaptation performance on new tasks (see Table1 in [3]). More recently, Anand et al [6] uses Metaworld with vision inputs, and shows that fine-tuning with vision inputs can adapt better than the meta-RL algorithms that use state-based inputs. Therefore, our work is built on and expanding these existing arguments, which, importantly, also include the recent literature in computer vision literature [7-10].
>
> In contrast to state-based environments which require many hand-designed observation inputs that could vary across tasks, vision-based observations are much easier to define, and provide a better opportunity for generalizing to new tasks: visual inputs not only side-step the discrepancy in dimensionally across different tasks, but may also offers common visual features that can be exploited and re-used for new tasks.
> Using a vision-based setup is also a must if we hope for any application of learning-based agents to real world tasks: the visual complexity is more challenging to learn and more closely resembles real world scenarios. Hence we believe that vision-based environments will likely be the dominating setup in future research.
>
>
>
>
> [1] Zhang, J., J. Wang, H. Hu, et al. Metacure: Meta reinforcement learning with empowerment-driven exploration. In International Conference on Machine Learning, pages 12600–12610. PMLR, 2021
> [2] Fakoor, R., P. Chaudhari, S. Soatto, et al. Meta-q-learning, 2020.
> [3] Xiong, Z., L. M. Zintgraf, J. Beck, et al. On the practical consistency of meta-reinforcement learning algorithms. ArXiv, abs/2112.00478, 2021.
> [4] Gao, K., O. Sener. Modeling and optimization trade-off in meta-learning. Advances in Neural Information Processing Systems, 33:11154–11165, 2020.
>
> [5] Yu, T., D. Quillen, Z. He, et al. Meta-world: A benchmark and evaluation for multi-task and meta reinforcement learning, 2021.
>
> [6] Anand, A., J. Walker, Y. Li, et al. Procedural generalization by planning with self-supervised world models. arXiv preprint arXiv:2111.01587, 2021.
>
> [7] Chen, W.-Y., Y.-C. Liu, Z. Kira, et al. A closer look at few-shot classification. Intl. Conference on Learning Representations, 2019.
>
> [8] Dhillon, G. S., P. Chaudhari, A. Ravichandran, et al. A baseline for few-shot image classification. Intl. Conference on Learning Representations, 2020.
>
> [9] Tian, Y., Y. Wang, D. Krishnan, et al. Rethinking few-shot image classification: a good embedding is all you need? In European Conference on Computer Vision, pages 266–282. Springer, 2020.
>
> [10] Chen, Y., Z. Liu, H. Xu, et al. Meta-baseline: exploring simple meta-learning for few-shot learning. In Intl. Conference on Computer Vision, pages 9062–9071. 2021.

---

> ### Comment · Reviewer_DrBc · 2022-08-08
> **Comments on Author response**
>
> Thanks for the clarification and I agree with the point about the vision-based input part. However, I still have some additional questions.
>
> Q2: I know there exist many algorithms in context-based Meta-RL using off-policy RL algorithms. What I mean here is that they are policy-based rather than value-based (SAC for pearl, PPO for varibad, TD3 for MQL). That is why I want to know if there exists any value-based context-based Meta-RL paper for previous literature. I believe it can work but it just lacks some evidences from previous literature. For gradient-based Meta-RL, things are completely different. I am sure it will make a large difference for MAML/reptile like algorithm when using value-based error (TD error) rather than policy-based error (policy gradient) as the inner/outer objective. One main reason is that the TD-error is not strictly related with the final RL performance (sometimes I can achieve low TD error but it does not mean my policy is good). The author needs to show there exist previous work that leverage TD error as the inner/outer objective for MAML-like algorithm to make it meta-RL baselines appropriate.
>
> Q3: I want to know how you implement your adaptation process. By (1) feed in all history trajectory into the context-encoder, for example, if you have steps like 1M, then basically the whole 1M transitions will be fed into the context-encoder, or (2) you will just feed in a few-trajectory like what normal meta-rl will do?
>
> Q5: Can you specify which plots or which part presenting the pretraining performance?

---

> > ### Author Response · Authors · 2022-08-09
> > **Author response to follow-up from reviewer DrBc**
> >
> > Thank you and we appreciate your follow-up response.
> >
> >
> > Q2 on prior work that uses TD-error for gradient-based meta-RL: thank you for the clarification and apologies for the confusion. We were not able to identify prior work that explicitly does so. However, we hope this concern can be addressed by noting 1) our results on Procgen are PPO-based and reflect similar results; 2) despite TD-error, gradient-based adaptation is able to success on other losses, such as image classification error, and behavior-cloning error [1]; 3) our results can be seen as a first attempt and useful reference if future work attempts at this direction.
> >
> >
> > Q3 on adaptation process: our implementation starts with (2), which feeds a few trajectories on the unseen task, but since the performance does not improve, we continue to do (1) and feed in longer trajectories. Thus the plotted performance corresponds to keep rolling out, and we remark that the 'normal' amount is a bit hard to define here, and running for longer is more as a sanity check.
> >
> > Q5 on Training results: please see the updated the submission which reports training performance for all methods on all benchmarks in Appendix C.2 "Training Results for Procgen Experiments"; Appendix D.2 "Training Results for RLBench Experiments" and Appendix E.2 "Training Results for Atari Experiments". Overall, the training performances are not significantly different across compared methods.
> >
> > [1] Cachet, T., J. Perez. Transformer-based meta-imitation learning for robotic manipulation. 2020.

---

### Official Review · Reviewer_qVWu · 2022-07-11

**Rating:** 6
**Confidence:** 4
**Soundness:** 4 excellent
**Presentation:** 3 good
**Contribution:** 2 fair

**Summary:**

The paper presents a study comparing popular meta-learning approaches like Reptile, Pearl and RL^2 with standard multi-task pretraining + fine-tuning on 3 vision based benchmarks, namely Procgen, RLBench and Atari. On all three benchmarks, they test the generalization ability of the approaches on a completely novel task rather than variations of existing tasks from the distribution. They show that on all the tasks multi-task pretraining with fine-tuning performs equally or better than the meta-RL counterparts proposing multi-task pretraining + fine-tuning as a simple yet strong baseline for such tasks.


**Questions:**

I would appreciate the authors making amends on either of the points mentioned in the weaknesses section.



**Limitations:**

The authors discuss the limitation of the RLBench and Atari benchmarks in terms of lack of diversity of the tasks and the resulting effect of the adaptation behavior of the algorithms. However, I would have liked to see some discussion along the lines of above mentioned grievances even if not presented in the paper.

**Strengths And Weaknesses:**

**Strengths**
1. The paper is well written and easy to read. I also like the precise claims in the paper, ( e.g, “our findings show that meta-learning approaches are evaluated on different tasks (rather than different variations of the same task), multi-task pre-training with fine-tuning on new tasks performs equally as well, or better, than meta-pretraining with meta test-time adaptation.”)
2. I also like the selection of tasks and the experimental setup as they cover 3 broad categories of vision based RL tasks.
**Weaknesses**
1. The paper shows that multi-task pretraining + fine-tuning perform equally/better than meta-RL approaches when presented with completely novel tasks rather than variations of existing tasks. I would have liked to view some results on variations of the same (vision based?) task as well to get a better sense of how much of the performance difference in the two scenarios.  Furthermore, there’s probably a spectrum of diversity where the scales start tipping from meta-RL to MT finetuning. I would be interested in understanding where it tips and the factors that determine it.
2. I would also like some additional baseline comparisons such as fine-tuning Pearl or RL^2 during adaptation. Currently the paper assumes the weights are fixed for these approaches during adaptation as in the original meta-learning setup. But the algorithm wasn’t designed to adapt to such completely novel tasks. So I think a fairer baseline to compare would be those algorithms with additional fine-tuning of their weights at test-time.

Overall, I think although the results in the paper aren't very surprising, but can serve as a documentation of the implicitly understood tradeoffs between meta-RL and Multitask training + finetuning.

---

> ### Author Response · Authors · 2022-08-02
> **Author response to reviewer qVWu**
>
> Thank you for providing detailed feedback and suggestions. We hope the following responses would address the raised questions and facilitate further discussions on how to improve our work.
>
> 1) Request for experiments on vision-based, multi-variation training:
>
> We added a new set of experiments to our updated supplementary material’s file (Figure 11). The task used is “push-button” from the RLBench task suite, where each variation corresponds to a different button color that the robot arm should push. Results show that all three compared methods can zero-shot perform the task, suggesting that the training task distribution, although being a finite discrete set of different colors, sufficiently covers the test-time color and allows generalization. However, we still haven't find a setup in the 3 used benchmarks that gives a stronger evidence for PEARL/RL^2 _gradually_ improving when being rolled-out in an unseen task. The only prior work to our knowledge is [3], where the RL^2 agent occasionally shows the expected adaptation behavior on a simplified version of the Coinrun environment (with the same color themes between train and test).
>
>
>
> 2) Request for investigation on the spectrum of diversity where the scales start tipping from meta-RL to MT fine-tuning.
>
> This is definitely an interesting question, but perhaps requires building more infrastructure on task suites that are better designed for this purpose than what we currently have in hand. Our current experiment still alluded to this question but didn’t elaborate in details: for example, the results in Figure 3 show that, as we expand the training set to 1000 levels, which is quite a large number of variations that’s comparable to the more traditional, state-based meta-RL benchmarks, the results between meta-RL and fine-tuning stay the same trend. On the other hand, in our Atari experiments, as we go to as low as 5 pretraing the tasks (Figure 7), the results also stay consistent with the overall conclusion. Hence our conclusion stays true across a decently wide spectrum of tasks, but we would love to hear back if the reviewer(s) have more suggestions in formulating further experiments in this direction.
>
>
> 3) Request for experiments on fine-tuning PEARL or RL^2:
>
> We expect that by allowing parameter updates, PEARL and RL2 would also improve on the test-time task and show a similar learning curve as their training performance. On tasks where zero-shot performance is already better than random (despite not improving afterwards), it’s also likely that fine-tuning these two methods could perform better than training from scratch. We are collecting more exact numbers for these results and will update on this thread once the results come together.
>
> In the meantime, there is also prior work on fine-tuning context-based meta-RL on state-based environments — [1] shows that PEARL can continue improving on the test-time task, and the adaptation procedure in [2] is essentially performing fine-tuning on context-conditioned policies.
>
>
>
>
> [1] Xiong, Z., L. M. Zintgraf, J. Beck, et al. On the practical consistency of meta-reinforcement learning algorithms. ArXiv, abs/2112.00478, 2021.
>
> [2] Zhao, Tony Z., et al. "Offline meta-reinforcement learning for industrial insertion." 2022 International Conference on Robotics and Automation (ICRA). IEEE, 2022.
>
> [3] Alver, S., D. Precup. A brief look at generalization in visual meta-reinforcement learning. arXiv preprint
> 445 arXiv:2006.07262, 2020.

---

> > ### Comment · Reviewer_qVWu · 2022-08-07
> > **Comments on Author response**
> >
> > I appreciate the author's response to my concerns and additional experiments being added by the authors. I'd be happy to increase the score to 6 once the authors add the results corresponding to the additional baselines they are trying!

---

> > > ### Author Response · Authors · 2022-08-09
> > > **Author response to reviewer qVWu**
> > >
> > > Thank you for taking the time for adding additional response. We updated the submission to reflect:
> > >
> > > 1) The requested, simper multi-variation setup, which is reported in Appendix D.3 and plotted in Figure 11. We remark that, on the “push button" task in RLBench, each method is trained on 10 variations and average test-time performance on 5 unseen variations. All compared methods can perform an unseen variation in a zero-shot manner, and PEARL is able to adapt despite the lack of gradient updates.
> > >
> > > 2) Training performance for all methods on all benchmarks, which are reported separately in Appendix C.2 "Training Results for Procgen Experiments"; Appendix D.2 "Training Results for RLBench Experiments" and Appendix E.2 "Training Results for Atari Experiments". Overall, the training performances are not significantly different across compared methods.
> > >
> > > 3) Results for finetuning context-based methods during adaptation. We report a set of experiments on finetuning PEARL on Atari games in Appendix E.4 "Finetuning PEARL-Rainbow Experiments" and Figure 17. On each game, we finetune the same PEARL-Rainbow agent for 10 runs, each corresponding to a different seed. We remark that, in contrast to rolling-out the agent (result in orange), finetuning (colored in green) PEARL can continue to improve on unseen game and match the finetuning results of mutli-task and Reptile-trained agents. This confirms prior work's finding that allowing gradient update to context-based meta-RL methods can indeed further improve their performance.

---

> > > > ### Comment · Reviewer_qVWu · 2022-08-09
> > > > **Thanks for the additional experiments**
> > > >
> > > > I thank the authors for the additional experiments performed. Bumped the score to 6

---

### Official Review · Reviewer_J2Vz · 2022-07-12

**Rating:** 4
**Confidence:** 4
**Soundness:** 3 good
**Presentation:** 3 good
**Contribution:** 2 fair

**Summary:**

This paper mainly questions the benefits of meta learning approaches in reinforcement learning by comparing them with multi-task learning (pretraining) with fine-tuning. Vision-based benchmarks, including Procgen, RLBench, and Atari, are used to investigate this research question. In most cases, a multi-task pretraining approach with fine-tuning peforms equally or better than meta-learning approaches, although it is simpler and computationally cheaper than meta-learning approaches.

**Questions:**

- could we do this similar investigation with some non-vision-based benchmarks? could image representation learning be the main bottleneck of learning in these benchmarks?
- in procgen experiments, why reptile is not working properly although it also fine-tunes the policy during adaptation as a multi-task pretraining approach does? (Other experiments show that reptile is similar to a multi-task pretraining approach)

**Limitations:**

As the authors mentioned in the paper,  the limited number of pre-training tasks for RLBench and Atari experiments are not appropriate for meta-RL, and therefore the results are not new and surprising. Moreover, the experiments are only conducted with vision-based environments, and therefore we could not easily generalize the results to other environments.

**Strengths And Weaknesses:**

Strengths
- Meta-RL approaches are fairly compared with a simple multi-task pretraining approach, and it shows that meta-RL is not the best option to leverage knowledge from previously learned tasks.


Weaknesses
- The authors only investigated several approaches with vision-based benchmarks. Their conclusion may only apply to vision-based settings.
- Meta-learning approaches generally requires a large number of tasks to be experienced during training, but the experiment settings don't have enough number of tasks. Therefore, the results are quite reasonable, but not surprising.
- The authors haven't provided any meaningful research direction to improve existing meta-RL approaches.

---

> ### Author Response · Authors · 2022-08-02
> **Author response to reviewer J2Vz - Pt2**
>
> 5) Question on why Reptile does not finetune properly on ProcGen.
>
> We apologize for the confusion, the reason is that we used a sub-optimal training setup for Reptile on ProcGen. In our reported results, fine-tuning Reptile yielded very small (but non-zero) performance improvement that may not be obvious from the plots.
>
> We went back and used a different training hyperparameter setup for Reptile, which led to better training performance than the previous version, and obtained better fine-tuning results that are much closer to results from fine-tuning the multi-task agent. The new results show a much more similar trend in results between ProcGen and RLBench — thank you for pointing this out and we hope the updated revision properly addresses this concern.
>
> [1] Xiong, Z., L. M. Zintgraf, J. Beck, et al. On the practical consistency of meta-reinforcement learning algorithms. ArXiv, abs/2112.00478, 2021.
>
> [2] Gao, K., O. Sener. Modeling and optimization trade-off in meta-learning. Advances in Neural Information Processing Systems, 33:11154–11165, 2020.
>
> [3] Yu, T., D. Quillen, Z. He, et al. Meta-world: A benchmark and evaluation for multi-task and meta reinforcement learning, 2021.
>
> [4] Anand, A., J. Walker, Y. Li, et al. Procedural generalization by planning with self-supervised world models. arXiv preprint arXiv:2111.01587, 2021.
>
> [5] Reed, Scott, et al. "A generalist agent." arXiv preprint arXiv:2205.06175 (2022).
>
> [6] Fan, Linxi, et al. "MineDojo: Building Open-Ended Embodied Agents with Internet-Scale Knowledge." arXiv preprint arXiv:2206.08853 (2022).

---

> ### Author Response · Authors · 2022-08-02
> **Author response to reviewer J2Vz - Pt1**
>
>
> Thank you so much for reviewing our submission and providing detailed feedback. We hope the following could address the raised concerns and questions.
>
> 1) Weakness due to using only vision-based environments.
>
> We’d like to emphasize that prior work has already done similar investigation on state-based environments, and shown results that are consistent with our conclusion. This comparison between finetuning and meta-RL has been done on the more ‘traditional’ meta-rl benchmark, i.e. multi-variation training and testing [1,2]. Another relevant work is the Metaworld benchmark [3], which provides multi-task with state-input settings, but MAML, RL2, PEARL are shown to give poor adaptation performance on new tasks (see Table1 in [3]). More recently, Anand et al [4] uses Metaworld with vision inputs, and shows that fine-tuning with vision inputs can adapt better than the meta-RL algorithms that use state-based inputs. Therefore, we believe that focusing on vision-based environments is a valuable setup, and instead of being limited, our conclusion should be well placed into the existing literature.
>
> Additionally, we believe that studying vision-based environments is very valuable and likely to be the dominating setup in future research. In contrast to state-based environments which require many hand-designed observation inputs that could vary across tasks, vision-based observations are much easier to define, and provide a better opportunity for generalizing to new tasks: visual inputs not only side-step the discrepancy in dimensionally across different tasks, but may also offers common visual features that can be exploited and re-used for new tasks. Using a vision-based setup is also a must if we hope for any application of learning-based agents to real world tasks: the visual complexity is more challenging to learn and more closely resembles real world scenarios.
>
> 2) Weakness due to not providing enough tasks to meta-RL algorithms
>
> We did provide enough number of tasks on ProcGen and RLBench: up to 1000 training levels are provided in our ProcGen environments, and we see consistent results despite varying the number of training levels from 100 to 1000 to 10000; on both RLBench and ProcGen, fine-tuning clearly outperforms training from scratch, suggesting the tasks setup are well suitable for meaningful adaptation to happen. On Atari, both finetuning and meta-RL underperform training from scratch, and this is where we concluded there is not enough overlap between train and test tasks to generalize. It’s indeed hard to say whether meta-RL algorithms would start out-performing if we scale the number of tasks to near infinite, and our result does not provide promising evidence for that.
>
> Moreover, prior work on state-based environments [1,2] provide the “enough” number tasks by using the more traditional multi-variation setup, but there’s still strong evidence that fine-tuning performs well (because the training task distribution is so limited, i.e. only varying one task parameter, [2] calls multi-task training ‘domain randomized search’). Hence we might conclude the reason for meta-RL methods under-performing fine-tuning should be about the algorithm and not the number of tasks.
>
> 3) Weakness due to not providing direction for improving meta-RL
>
> This is not in the scope of our work, and our results actually suggest it might be more promising to shift from meta-RL to fine-tuning. We have focused and put in much effort in investigating which line of approach is more effective, hence we hope to bring interesting insights and pose the direction for future research as an open-ended question.
>
> Since our submission, instead of improvement on meta-RL algorithms, new work [5] has actually shown promising generalization results on fine-tuning imitation-learning agents, which suggests that efforts on scaling up task complexity, also a main theme in [6], and using simpler, more stable approaches might be the more promising direction in the future.
>
> 4) Question on whether image representation learning is the main bottleneck of learning
>
> We don’t believe so. On all 3 benchmarks, all results from multi-task and training-from-scratch showed successful learning of completing the task(s) from vision-only inputs. The base RL algorithms (PPO for ProcGen, C2F-ARM for RLBench, RainbowDQN for Atari) have also been previously proven effective at solving these benchmarks. Notably, the Atari game environments provide the least challenge for representation learning: each task has a very small image observation space (single-channel, 84x84 in size) and requires small convolutional networks for image encoding as compared to the Q-attention networks used for C2FARM in RLBench.

---

### Meta-Review · Area_Chair_jGyL · 2022-08-26

**Recommendation:** Accept
**Confidence:** Certain

**Metareview:**

Reviewer qvwu summarizes the paper well: The paper presents a study comparing popular meta-learning approaches like Reptile, Pearl and RL^2 with standard multi-task pretraining + fine-tuning on 3 vision based benchmarks, namely Procgen, RLBench and Atari. On all three benchmarks, they test the generalization ability of the approaches on a completely novel task rather than variations of existing tasks from the distribution. They show that on all the tasks multi-task pretraining with fine-tuning performs equally or better than the meta-RL counterparts proposing multi-task pretraining + fine-tuning as a simple yet strong baseline for such tasks.

The other reviewers voted to reject the paper. Their main concerns were:
- Evaluation of vision-only benchmarks
- Issues in evaluation setup

I believe the authors have sufficiently addressed these concerns, but unfortunately, the reviewers did not respond to the authors. In particular prior work already shows fine-tuning is competitive to meta-learning algorithms on state-based tasks. The main contribution of this work is in showing that these findings hold true in vision-based settings too and particularly in the scenario where tasks are different. This is a useful addition to the growing body of work comparing vanilla fine-tuning against meta-learning. I therefore recommend the paper be accepted.

**Award:**

No

---

### Decision · Program_Chairs · 2022-09-14

Accept